# Area level impacts on emergency hospital admissions of the integrated care and support pioneer programme in England: difference-in-differences analysis

Eilís Keeble,[1] M Bardsley,[1] Mary Alison Durand,[2] Ties Hoomans,[2] Nicholas Mays[2]

¹The Nuffield Trust, London, UK
²Policy Innovation and Evaluation Research Unit, Department of Health Services Research and Policy, London School of Hygiene and Tropical Medicine, London, UK

**Correspondence to**
Eilís Keeble;
eilis.keeble@nuffieldtrust.org.uk

## ABSTRACT

**Objective** To examine whether any differential change in emergency admissions could be attributed to integrated care by comparing pioneer and non-pioneer populations from a pre-pioneer baseline period (April 2010 to March 2013) over two follow-up periods: to 2014/2015 and to 2015/2016.

**Design** Difference-in-differences analysis of emergency hospital admissions from English Hospital Episode Statistics.

**Setting** Local authorities in England classified as either pioneer or non-pioneer.

**Participants** Emergency admissions to all NHS hospitals in England with local authority determined by area of residence of the patient.

**Intervention** Wave 1 of the integrated care and support pioneer programme announced in November 2013.

**Primary outcome measure** Change in hospital emergency admissions.

**Results** The increase in the pioneer emergency admission rate from baseline to 2014/2015 was smaller at 1.93% and significantly different from that of the non-pioneers at 4.84% (p=0.0379). The increase in the pioneer emergency admission rate from baseline to 2015/2016 was again smaller than for the non-pioneers but the difference was not statistically significant (p=0.1879).

**Conclusions** It is ambitious to expect unequivocal changes in a high level and indirect indicator of health and social care integration such as emergency hospital admissions to arise as a result of the changes in local health and social care provision across organisations brought about by the pioneers in their early years. We should treat any sign that the pioneers have had such an impact with caution. Nevertheless, there does seem to be an indication from the current analysis that there were some changes in hospital use associated with the first year of pioneer status that are worthy of further exploration.

### Strengths and limitations of this study

► This study adds to the evidence of the impact of system-wide approaches to integrating health and social care, like the integrated care and support pioneer programme, using advanced statistical methods to determine whether the pioneers reduced emergency admissions.

► Reducing emergency admissions is often cited as a key goal of new integrated models of care and the Hospital Episode Statistics provide a continuously collected person level dataset to enable tracking of changes over time at small area level.

► Analysing the pioneer sites collectively ensured the inclusion of a diverse range of areas which were unlikely to be systematically different at baseline from the non-pioneers.

► It is difficult to find a true counter-factual population to compare with the pioneers as many other initiatives related to health and social care integration had been developed in other areas of the country previously and/or were being implemented almost simultaneously.

► The pioneers invested in a collection of health and social care integration strategies and interventions; identifying the causes and effects of these specific initiatives would require detailed local primary data collection but this analysis focuses on the overall impact of the pioneers as a national policy initiative.

## INTRODUCTION

In November 2013, the integrated care and support pioneer programme was initiated in England. The programme aimed to promote integration between the separate local health and social care systems in England by facilitating these systems to develop and implement new ways of working together with the objective of meeting people's needs better and improving service users' experience of care.[1]

In the first wave of the programme, the English Department of Health (now Department of Health and Social Care) selected 14 pioneer areas from a round of competitive applications, that were identified as the 'most ambitious and visionary' in their plans for health and social care system integration.[2]

Each pioneer was given access to limited support and expertise over a 5-year period and a one-off fund of £90 000 to help with initial development. A second wave of 11 pioneer areas was subsequently announced in January 2015. These are excluded from the present analysis as there are insufficient time points available currently for an interpretable trend analysis.

Integration in the pioneer areas has taken on different forms. As Erens and colleagues noted, 'What it meant to be 'a pioneer' varied between sites and between individuals within sites. At various times, it was apparent that pioneer status meant one or more of the following:

1. A 'badge' for a locality signifying national recognition of innovation and progress in integrating care.
2. An enabler of the existing local plan for transformation.
3. A particular governance arrangement, for example, a board that brought all system leaders and their organisations around the table.
4. A collection of discrete workstreams, characteristically covering a combination of different groups of users and infrastructure projects (eg, information sharing, workforce development and so on).
5. A specific new integrated service, such as a frailty service.
6. An ethos or way of thinking about and providing care, rather than a specific plan or set of initiatives'.[3]

Some of the pioneers planned to focus on specific populations. Of these, the most common were older people, people with long-term conditions and people at high risk of hospitalisation. Broadly, however, the pioneers shared the same vision for the future of the health and social care system by seeking to create a 'whole system' of integrated care involving all local bodies and professional groups organised around the needs of individuals and their informal carers which set them apart from the rest of England.[4]

All but one (Stoke and North Staffordshire) of the wave 1 pioneers stated that reducing emergency admissions was an aim or an expected outcome of integration in their original twice daily. Risk stratification with targeted interventions and introducing preventive strategies to avoid the need for acute hospitalisation were listed as activities to achieve this goal (see online supplementary material). The focus on reducing emergency hospital care use was given still greater emphasis by the pioneers as financial austerity bit more deeply into local healthcare budgets after 2013.[3]

As a consequence of the focus on emergency hospital care as a costly service, the success of integrated care initiatives has often been presented, at least in part, in terms of their ability to reduce the need for emergency hospital admissions and to reduce emergency admission rates.[5] Reducing emergency admission rates has been a feature of English health policy over the last decade and continues to be one of the most commonly used measures of success for system change initiatives.[6–8] To date, however, there has been little evidence of initiatives successfully reducing emergency admissions.[9–11]

This paper presents new evidence on the effect of the pioneer programme on emergency admissions. We investigate changes in the emergency admissions to hospitals of patients across England following the implementation of the programme in 2013. The analysis is part of a wider programme of evaluation of the pioneers (http://piru.lshtm.ac.uk/projects/current-projects/integrated-care-pioneers-evaluation.html). Though it is not possible to identify precisely which elements of the programme, if any, led to any differential change observed (since the pioneers were not working from an agreed template), such an analysis can be justified as a necessary step in understanding the impacts of a major initiative such as the integrated care and support pioneer programme, especially since it had much in common with successive initiatives such as the New Care Model Vanguards and the current focus on integrated care systems.[12 13] The underlying hypothesis is that the cumulative effect of the specific initiatives embedded in each pioneer programme would bring about sufficient change in emergency hospital care use as to be detectable at the level of the whole population of the pioneers.

## METHODS
To examine whether differential change in emergency admissions could be attributed to pioneer status, we used a difference-in-differences approach. Difference-in-differences measures the effect of the intervention (the pioneer programme) by looking at the change in emergency admissions between the preintervention and postintervention periods in the two groups and quantifies whether or not the population within the pioneer programme experiences a change that is significantly different to the comparison group, the non-pioneers.

### Data sources
We used inpatient Hospital Episode Statistics (HES) to identify all emergency admissions to NHS hospitals in pioneer and non-pioneer areas across England. HES is collated by NHS Digital and is a pseudonymous patient level dataset that records basic features of admissions to hospital including patient age, sex, admission date and an emergency admission indicator (admission methods starting with '2').[14]

To be able to compare emergency admission rates between areas (pioneer/non-pioneer), we also obtained information on key local authority level factors determining local population health and care needs:

1. Demographic composition (age and sex), from the Office for National Statistics.[15]
2. Deprivation decile, from the 2015 Index of Multiple Deprivation.[16]

### Defining pioneer areas
The pioneer areas did not all map neatly to a single set of health or local government administrative boundaries. After consultation with each pioneer, they were mapped

to the local authorities which most closely aligned with the intervention area (see online supplementary material for lookup table). Local authority boundaries were used instead of health boundaries as the population denominators could be linked over a longer period. A wider breadth of data is available for this boundary which is being used in other parts of the evaluation, for example, social care data.

The local authorities which were linked to the second wave of pioneers, initiated in January 2015, were excluded from all analyses and not included in either the pioneer or non-pioneer populations. Non-pioneer areas were defined as any local authority that was not a first or second wave pioneer.

### Defining time periods

A baseline period before pioneer programme implementation of April 2010 to March 2013 was compared with two follow-up periods: April 2014 to March 2015 (2014/2015) and April 2015 to March 2016 (2015/2016). The period April 2013 to March 2014 was excluded as this encompassed the call for applications to the programme (May 2013) and the announcement of the sites (November 2013).

### Outcome

Our primary outcome was the average percentage difference in rates of emergency hospital admissions per 100 000 between baseline and follow-up (2014/2015 or 2015/2016) for the study groups (pioneers/non-pioneers). Area-level rates were calculated as the total number of emergency admissions over each time period divided by the mid-year population for each group. Admissions were derived by month and local authority of residence. They were adjusted for deprivation decile, age group (0–19, 20–39, 40–59, 60–79, 80+ years) and sex. The English age, sex and deprivation decile structure were used as the reference population for each local authority for the initial analysis. The secondary outcome was the difference in average percentage change in the rates over time between the pioneers and non-pioneers.

### Statistical analyses

An initial difference-in-differences comparison was performed by looking at the change in the adjusted emergency admission rate for the pioneers and non-pioneers. Percentage differences between the baseline period and the two follow-up time points of 2014/2015 and 2015/2016 were calculated, along with the difference between these.

To determine whether the change in emergency admissions in the pioneers was significantly different from the change in the non-pioneers, we performed difference-in-differences regression analysis. We estimated negative binomial regression models for count data adjusting for age, sex and deprivation decile. Poisson models were first attempted but the data were over-dispersed and unsuitable. Each regression model included a continuous local authority population size exposure variable, a binary pioneer status term (pioneer/non-pioneer), a binary time term (baseline/follow-up), a difference-in-differences term (pioneer status*time) and covariate terms. We obtained robust SE estimates adjusting for clustering of the repeated measures from each local authority. Significance was assessed at p<0.05. SAS V.9.4 was used for all analyses.

### Difference-in-differences estimation validation tests

To validate our difference-in-difference estimations, we tested the following assumptions:

1. That areas were not selected into the programme based on emergency admission rates at baseline, by comparing baseline emergency admissions and demographics of the pioneer and non-pioneers.
2. That changes in emergency admission rates over time would be the same for both the pioneer and non-pioneer areas in the absence of the pioneer programme, by comparing adjusted emergency admission rates for the pioneers and non-pioneers over the baseline period. These were compared graphically and statistically using a linear time trend of month in the baseline period interacted with pioneer status controlling for age, sex and deprivation decile.

### Sensitivity analyses

We examined sensitivity of the main findings to excluding Stoke and North Staffordshire Pioneer from our analyses and to using individual years for the baseline period (see online supplementary material). Stoke and North Staffordshire had a unique target population and no focus on reducing emergency admissions. As the baseline period covered 3 years, each individual baseline year was also compared with the first follow-up time point.

### Patient and public involvement

Patient and public representatives are involved in the wider evaluation of which this analysis forms a part and were involved in the selection and peer review of the initial proposal on which this analysis is based.

## RESULTS
### Baseline characteristics

The characteristics of the pioneers and non-pioneers during the baseline period of April 2010 to March 2013 are summarised in table 1. The pioneers consisted of 49 local authorities and encompassed 17% of the English population in the baseline period.[15] The proportions of the population aged 65 years and over, or female, were similar between the two groups. Area level deprivation in the pioneers was slightly higher than in the non-pioneers.

### Trend analysis

Figure 1 shows the adjusted monthly emergency admission rates for the pioneers and non-pioneers between April 2010 and March 2016. On visual inspection, the trends in the baseline period overlap which indicates

**Table 1** Baseline characteristics of the pioneer and non-pioneer populations

| Characteristic | Pioneers first wave (n=14)* | Non-pioneers |
| --- | --- | --- |
| No of local authorities | 49 | 244 |
| Average yearly population at baseline | 9 083 051 | 37 137 613 |
| Proportion population under 20 | 24% | 24% |
| Proportion population aged 65+ | 16% | 17% |
| Proportion population female | 50% | 50% |
| Average local authority IMD score (2015) | 21.1 | 18.7 |

*11 s wave pioneers and 33 associated local authorities were excluded from the analyses.

that trend bias should have limited impact on the difference-in-differences analysis. A statistical test of the trends in the baseline period also indicated limited trend bias (p=0.7156).

### Difference-in-differences

Between the baseline period and the first follow-up period (2014/2015), average emergency admission rates decreased by 0.42% for the pioneers and increased by 3.46% for the non-pioneers, with a difference-in-differences of 3.89% (see table 2). When the baseline was compared with the second follow-up period (2015/2016), the pioneers still had a lower increase at 2.23% but the difference compared with the non-pioneers was smaller at 3.23%.

Trends for the individual pioneers varied. For example, half the pioneers had a percentage increase in their emergency admission rates between baseline and 2014/2015, while the percentage difference for the pioneers as a whole was a slight decrease (see table 3). There was also variation within pioneers (see online supplementary material). For example, the constituent local authorities comprising the Waltham Forest, East London and City Pioneer had declines in emergency admission rates ranging from −10.45% (Tower Hamlets) to −1.64% (Newham) between baseline and 2014/2015, while the overall percentage difference was −5.73%.

### Difference-in-differences regression

After adjusting for age, sex and deprivation, the difference-in-differences regression analysis showed that the change in emergency admission rates in the pioneers between baseline and 2014/2015 was smaller and significantly different from that of the non-pioneers (p=0.0379)

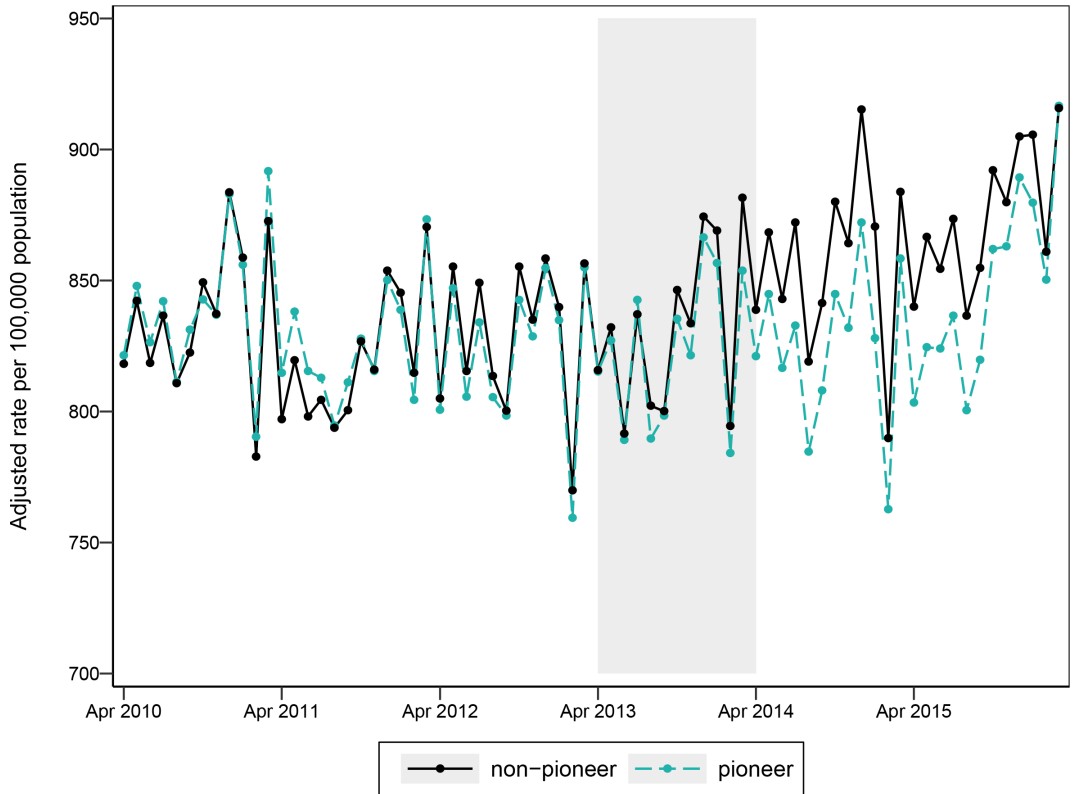

**Figure 1** Emergency admission rate for pioneers and non-pioneers by month adjusted for age, sex and deprivation decile (pioneer intervention introduced in shaded area).

**Table 2** Emergency admission rates for pioneers and non-pioneers (adjusted for age, sex and deprivation decile) at baseline and follow-up, with percentage differences compared with baseline and difference-in-differences between non-pioneers and pioneers

| | Emergency admission rate (per 100 000 population) | | | Percentage difference | | Difference-in-differences* | |
| --- | --- | --- | --- | --- | --- | --- | --- |
| | Baseline | 2014/2015 | 2015/2016 | 2014/2015 | 2015/2016 | 2014/2015 | 2015/2016 |
| Non-pioneer | 9942 | 10287 | 10485 | 3.46% | 5.46% | 3.89% | 3.23% |
| Pioneer | 9948 | 9906 | 10170 | −0.42% | 2.23% | | |

*Difference between the non-pioneer and pioneer percentage differences, positive value indicates non-pioneer change is greater.

(see table 4). The pioneer emergency admission rate increased by 1.93% compared with 4.84% in the non-pioneers. When comparing baseline with 2015/2016, the analysis still indicated that the change in emergency admissions for the pioneers was smaller at 4.01% compared with 6.33% for the non-pioneers but the difference was not statistically significant (p=0.1879).

### Sensitivity analyses

Excluding Stoke and North Staffordshire did not affect the overall findings but reduced the significance of the difference between the pioneers and non-pioneers in 2014/2015 (p=0.0464); however, this exclusion also meant the trends were less parallel and subject to more bias from the baseline period (p=0.3030). After comparing individual baseline years to 2014/2015, all years found a smaller change for the non-pioneers but only 2012/2013 was statistically significant (p=0.0189), this was also the baseline year with the most parallel trends for pioneers and non-pioneers (p=0.9899). Full results presented in online supplementary material.

### DISCUSSION

The integrated care and support pioneers represent one important example of how English health and social care services have been exploring new ways of working across organisational boundaries. The aims of the individual pioneers varied,[3] but most had a common interest in providing care and support that was intended to reduce the need for urgent care services and lead to a reduction in emergency hospital admissions. After comparing changes in emergency admissions from a 3-year pre-pioneer baseline period between pioneer and non-pioneer populations, we found a lower increase in emergency admissions for the pioneers than the non-pioneers. This lower increase was statistically significant for the comparison between baseline and 2014/2015 (p=0.0379) but not for the comparison between baseline and 2015/2016 (p=0.1879).

This type of population level analysis can help provide some independent evidence of the likely scale of changes within an area associated with integrated care initiatives

**Table 3** Emergency admission rates for individual pioneers (adjusted for age, sex and deprivation decile) at baseline and follow-up, with percentage differences compared with baseline

| | Emergency admission rate (per 100 000 population) | | | Percentage difference to baseline | |
| --- | --- | --- | --- | --- | --- |
| Pioneer (no of Local Authorities) | Baseline | 2014/2015 | 2015/2016 | 2014/2015 | 2015/2016 |
| Barnsley (1) | 10992 | 11769 | 12325 | 7.07% | 12.13% |
| Cheshire (2) | 11259 | 12160 | 12459 | 8.00% | 10.65% |
| Cornwall and Isles of Scilly (2) | 8170 | 8061 | 8193 | −1.33% | 0.29% |
| Greenwich (1) | 8168 | 8226 | 9513 | 0.71% | 16.47% |
| Islington (1) | 6324 | 6320 | 6096 | −0.06% | −3.60% |
| Kent (12) | 9349 | 10033 | 10009 | 7.32% | 7.06% |
| Leeds (1) | 11399 | 9605 | 10155 | −15.74% | −10.91% |
| North West London (8) | 8922 | 8665 | 8812 | −2.87% | −1.23% |
| South Devon and Torbay (3) | 7415 | 7630 | 8803 | 2.90% | 18.72% |
| South Tyneside (1) | 11153 | 10445 | 11150 | −6.35% | −0.03% |
| Southend (1) | 9243 | 10397 | 10224 | 12.49% | 10.61% |
| Stoke and North Staffordshire (7) | 9949 | 10253 | 10611 | 3.06% | 6.66% |
| Waltham Forest, East London and City (3) | 9184 | 8657 | 8279 | −5.73% | −9.85% |
| Worcestershire (6) | 9018 | 8817 | 9006 | −2.23% | −0.13% |

**Table 4** Difference in difference model coefficients and percentage difference in emergency admissions for pioneers and non-pioneers, adjusted for age, sex and deprivation

| | 2014/2015 | 2015/2016 |
|---|---|---|
| **Model coefficients (p value)** | | |
| Intercept | −5.4207 (<0.0001) | −5.4231 (<0.0001) |
| Non-pioneer/pioneer | −0.0060 (0.7524) | −0.0060 (0.7537) |
| Baseline/follow-up | 0.0473 (<0.0001) | 0.0614 (<0.0001) |
| Interaction | −0.0282 (0.0379) | −0.0221 (0.1879) |
| **Percentage difference (95% CI)** | | |
| Non-pioneer | 4.84 (3.67 to 6.03) | 6.33 (5.00 to 7.68) |
| Pioneer | 1.98 (−0.43 to 4.34) | 4.01 (0.95 to 7.16) |

and curb some of the more zealous rhetoric for or against integrated health and social care, and the related changes in service delivery. Looking at emergency admission data on this scale means the outcome of interest is based on a relatively large number of events and continuously collected data—making them useful as a measure of potential programme impact. This is in contrast to a range of other potential measures of health and social care integration at community level that are likely to be less sensitive to short-term change such as annual patient experience surveys. The size and range of geographical areas covered by both the pioneers and non-pioneers along with their sociodemographic similarities should mean that differences in factors such as supply of social care services or acute hospital beds and the process of collecting data are unlikely to be systematically different between the two groups beyond any changes associated with pioneer status.

It would be beneficial to track emergency admissions for >2 years to measure the impact of policy initiatives such as the pioneers more definitively. However, during the life of the pioneer programme, there were parallel changes in the wider policy context both in terms of specific health and care integration policies such as the Better Care Fund,[17] the overall level of funding for both health and social care in a period of unprecedented financial austerity and,[18] from 2015 onwards, the New Care Model Vanguards.[19] In particular, the Vanguards' approach to improving care coordination had much in common with the pioneers. This means that the ideas behind integration that prompted the pioneers and the types of interventions that they developed are no longer (if they ever were) unique to these areas and are being implemented across the country. Therefore, a true counter-factual population is difficult to find. This may, in part, explain why the difference between the pioneers and non-pioneers reduced between baseline and 2015/2016 compared with baseline and 2014/2015 as the behaviour of the non-pioneers becomes increasingly similar to the pioneers.[20] This is in part to be expected as disseminating learning from the pioneers was actively encouraged as part of the programme.

In addition to the difficulties of finding a counterfactual over the life-time of the pioneers, it was not the first programme to focus on health and social care integration in England. One such previous initiative was the Integrated Care Pilots. While an effect on emergency admissions was not found for this programme, it cannot be ruled out that these pilots have had a legacy impact on emergency admissions.[21] It should therefore be noted that, three of the pioneers overlap with areas that were previously Integrated Care Pilots (Cornwall, Torbay and Tower Hamlets) and therefore, may have had a focus on integration for longer than some other pioneers. This may in part explain the steady declines in emergency admissions seen in Tower Hamlets and to a lesser extent Cornwall. Seven of the Integrated Care Pilots also covered areas which were not pioneers and therefore, the impact of the pioneers in contrast to these may be reduced.

A more detailed understanding of the impacts of the pioneers would be gained with a targeted analytical approach using information on the specific initiatives implemented in each pioneer and data on the exact populations in receipt of these initiatives (this is being attempted in another component of the pioneer evaluation). While this might yield gains in terms of causal inference in that changes could potentially be attributed to a specific set of local actions, such an analysis might lose the ability to assess the impact of change across a system and an entire population. This is important to note as the pioneers were intended to be a complex mix of specific service changes and initiatives, supported by a wider pattern of infrastructural changes at the level of the local health and social care system.

Other studies have looked at schemes with an aspiration to reduce the need for urgent hospital care through better coordinated health and care services, and with an emphasis on preventing admissions. Success is typically assessed in terms of reduction in emergency hospital admissions and various previous evaluations show that this has been difficult to achieve.[9–11] Despite the intense policy interest in how different forms of service delivery can reduce emergency admissions, there are few, if any, studies showing unequivocal change in the direction desired. Against this backdrop, the modest changes observed across the 14 wave 1 pioneer areas in their first 2 years look promising. However, when exploring the extent to which the observed changes are likely to be related causally to pioneer status, it should be noted that:

1. The effect appears to be temporary: and as such the effect may have been linked to changes that took place in the early stages of the pioneers or pre-pioneer but were not sustained; or the non-pioneer areas introduced changes which have subsequently reduced the difference between them and the pioneers.
2. The changes in emergency admissions were not shown in all places and even varied between local authority areas within the same pioneer.

## CONCLUSION

It is ambitious to expect unequivocal changes in a single high level and indirect indicator of health and care integration such as emergency hospital admissions to arise as a result of changes in local health and care provision across organisations brought about by the pioneers in their early years. We should treat any signs that the pioneers have had such an impact with caution. Nevertheless, our analysis does seem to provide some evidence that there were some changes in hospital use associated with the first year of pioneer status that are worthy of further exploration. At the very least, this analysis shows that pioneer status does not seem to have been associated with a relative deterioration in performance in terms of emergency hospital use.

**Contributors** EK undertook the analysis and with MB drafted the initial paper. MB and NM contributed to design of the study. EK, MB, MAD, TH and NM contributed to interpretation of findings and revisions of the paper.

**Funding** This study is funded by the NIHR Policy Research Programme (Evaluation of the Integrated Care and Support Pioneers Programme in the context of new funding arrangements for integrated care in England (2015-2020), PR-R10-0514-25001).

**Disclaimer** The views expressed are those of the authors and are not necessarily those of the NIHR or the Department of Health and Social Care.

**Competing interests** None declared.

**Patient consent for publication** Not required.

**Provenance and peer review** Not commissioned; externally peer reviewed.

**Data availability statement** This study used Hospital Episode Statistics data obtained from NHS Digital under a data sharing agreement and are reused with their permission. Hospital Episode Statistics data may be obtained from NHS Digital under a similar process but we are unable to share it per the terms of our agreement.

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
