## [Reviewer comments · BMJ Open]

ARTICLE DETAILS

TITLE (PROVISIONAL)	Area level impacts on emergency hospital admissions of the Integrated Care and Support Pioneer Programme in England: difference-in-differences analysis
AUTHORS	Keeble, Eilís; Bardsley, M; Durand, Mary Alison; Hoomans, Ties; Mays, Nicholas

VERSION 1 – REVIEW

REVIEWER	Lu Han Research Fellow Department of Health Sciences University of York UK
REVIEW RETURNED	06-Dec-2018

GENERAL COMMENTS	This manuscript aimed to evaluate the impact of the Integrated Care and Support Pioneer programme in England, with 14 areas being selected for the initial wave in November 2013. The primary outcome being evaluated was emergency hospital admissions. This is a complex health policy as the 14 pioneer areas were not randomly selected and were being able to design and deliver integrated care based on their local health needs. Although the 14 areas broadly shared a common goal aiming to improve the quality of care and patients' experience by creating a 'whole system', I would imagine there existed great variations in terms of the healthcare organisation and delivery. The evaluation of this health policy, therefore, would require careful design to take into account the heterogeneity not only among pioneer areas but also among pioneer and non-pioneer areas, in order to separate the impact of interest from other confounding factors. Major points: 1 I think a better understanding on the similarities and differences in the pioneer plan among the 14 areas would be helpful for the audiences to see the whole picture. As the authors stated, many of the pioneers planned to focus on older people, people with long-term conditions, and families with complex health needs. Other pioneers would put their efforts on reducing the use for hospital care and therefore to ease the economic burden of the local healthcare system. A summary table comparing the key concerns of each pioneer area and the policies implemented to address these issues would be helpful.
--

	2 This is related to point 1. After the area-level pioneer plan has been identified, a theoretical framework could be included in the introduction to demonstrate the mechanism through which the outcome (emergency admission rate) is related to which element of the plan. This would help justify the validity of the research question. 3 Study design. I understand the challenges that one would face when evaluate the impact of a complex health policy, and the DiD analysis could be an effective method. However, the key assumption for DiD analysis to be able to isolate the effect of interest is that the intervention and control units have balanced characteristics in other aspects that would affect the outcome. Therefore, it might be useful to identify matched control areas for each pioneer area based on the factors that would potentially affect people's health needs, and compare the pioneer areas only to the matched control areas (rather than the rest of the country). 4 This is related to point 3. As the study is using HES data, I wonder if the analyses can be applied on hospital-level or even patient-level data, based on the location of hospitals. Then more potential influential characteristics could be controlled for. 5 This study compared the baseline emergency hospital admission rate with this outcome in 2014/15 and 2015/16, the years immediately after the implementation of the pioneer plan. I wonder if this is a short period for the pioneer plan to show a detectable impact, if there is any. How quickly an impact can be observed would be depending on the focus and delivery of the plan. For example, children with asthma might benefit from the integrated care with reduced risk to be admitted to hospital as emergencies sooner than older population with long-term conditions, for whom we might need longer time to observe a reduced need for emergency hospital care. Therefore, the pre- and post- differences in the outcome as detected in this study might not be taken as evidence of a significant association between the pioneer programme and the use of emergency hospital care. Minor points:  1. Figure 1, as mentioned on page 8 line 26, seems to be missing from the manuscript. 2. It would be useful to include more information on the comparison between pioneer and non-pioneer areas for the pre-intervention trend.
--	--

REVIEWER	Martin Roland University of Cambridge UK
REVIEW RETURNED	12-Dec-2018

GENERAL COMMENTS	A number of interventions in recent years, especially in the UK, have sought to reduce emergency hospital admissions which are seen as a potential way of reducing health service costs / utilisation without necessarily having a negative impact on population health. The authors take a conventional approach to analysing the available observational data, given the absence of randomisation.
--

	For Pioneer groups, the authors state that 'a number aimed to reduce reliance on emergency hospital care'. How many of the 49 had this aim, and should the analysis have been restricted to those which had reducing admissions as an explicit aim? The later focus on reducing emergency hospital use 'as financial austerity bit more deeply' may have come too late for the current analysis (making this group more like second wave Pioneers whom the authors excluded from the analysis). What were the other ones trying to do (granted that the over-arching aim was described as integration)? In general the paper could say more at the start more about what the Pioneers were trying to do (details are given but not till the discussion section) as international readers will have no idea what a 'Pioneer' or 'non-Pioneer' is. It would be helpful to know how the non-Pioneer groups were selected. Were these all areas in England not defined as Pioneers? This is not clear. The Pioneer programme was not the first in England to promote integration and some previous ones had claimed successes in reducing admissions. Were these excluded from the controls? This would at least be worth a mention in the discussion. In the analysis, an initial finding is that the intervention group had a 3% greater rate of emergency admissions at baseline (table 1, 10,013 vs 9,705). This raises the possibility that the marginal effect which the authors found in terms of reduced rate of rise in admissions in the intervention group could represent regression to the mean (i.e. if you randomly selected a group with higher rates at baseline, they're likely increase less). Rather than using a method that would have used all the data (e.g. segmented regression), the authors have chosen to average the rates in the pre-intervention period and then compare this average with subsequent changes. This seems to depend on their observation that 'the trends were close to parallel' (Results, Trend analysis, line 2) – they may be parallel but they're certainly not flat as, eyeballing figure 1 it's clear that admissions were increasing more rapidly in the Pioneers than non-Pioneers. A statistical opinion on the analytic method chosen should be sought. Other than this, I found the paper well written with judiciously cautious conclusions. One point that could have been made is that some similar evaluations find that the effects of this type of complex intervention take some time to become evident (e.g. ~2 years). Here the reverse seems to be the case, with the effect waning in the second analytic period, though this could have been due to contamination of the control group by other NHS initiatives to reduce admissions.
--	---

REVIEWER	Marcello Morciano University of Manchester, UK
REVIEW RETURNED	12-Dec-2018

GENERAL COMMENTS	The manuscript assesses the impact estimates of the Wave 1 of the Integrated Care and Support Pioneer Programme. This is an observational study based around a standard difference-in-difference framework. The authors tested whether the introduction of the programme was associated with reduced emergency hospital admissions. Non-pioneer sites experience a significant
--

	increase in emergency admissions. These have grown more slowly in pioneer sites. The subject matter is topical, and the issue of robustness and validity of impact estimates a tricky one. There are several aspects of the analysis that prevent me to endorse the manuscript's acceptance at present stage.  - Pioneer sites were mapped into Local Authorities. I do not find the justification of using local authorities boundaries (instead e.g. CCG boundaries) very compelling. Additionally, CCGs were created in 2013, creating a very serious confounding for the analysis, because of the overlapping with the Pioneer Wave 1. - Is there any specific reason of collapsing data at year level (and not e.g. at monthly level)? [line 20,p. 6]. - Standardisation: "For the rates, the emergency admissions were directly standardised using the age and sex structure of the 2015 mid-year population estimates for England" [lines 29-33 p 6]. What "directly" means in this context is unclear? Could the authors clarify whether rates were standardised according population estimates for the entire England or according local authority level population estimates? - I've found the (presentation of the) statistical analysis lacking. In particular:  a. Did the author test formally for trend parallelism? "the trend seem close to parallel" [p.8 line 28] is a vague statement. b. What is the rationale of using a negative binomial regression model when results are discussed exclusively in terms of emergency admission rates? c. "To model the emergency admissions counts as a rate, an offset of the log of the population size in each local authority was incorporated into the model." This is not the standard approach? Were emergency admissions log-scaled? d. I understand "time" takes only 4 values: 1 (2010/11),...,4 (2014/15). Could the authors clarify the reason behind "A baseline of the average of the emergency admissions in the financial years 2010/11 to 2012/13 was created as the period before the initiation of the Pioneer programme in late 2013"? how time enter in the regression analysis? Moreover, Figure 1 reports outcomes from 2007/8, whereas I understand that the statistical analysis uses the "baseline period". Could you also clarify the reasons behind that choice? e. "The model accounted for the repeated measures from each local authority and was adjusted for population age group, sex and area-level deprivation" [lines 27-28 p. 7] It is not clear to me whether this would imply the use of a fixed-effect model or instead local authorities dummies were included in the model. Is area-level deprivation a time-invariant characteristic? I am puzzling whether your model is fully identified. f. Do you include population size as a regressor in the Poisson specification? g. Table 4: what "[..]" "calculated" percentage difference" would mean? h. I have found the Discussion lacking. I wonder whether the authors could complement the quantitative analysis with some qualitative evidences.
--	--

VERSION 1 – AUTHOR RESPONSE

Point	Reviewer	Location	Comment	Proposed Solution
1	1	Introduction	I think a better understanding on the similarities and differences in the pioneer plan among the 14 areas would be helpful for the audiences to see the whole picture. As the authors stated, many of the pioneers planned to focus on older people, people with long-term conditions, and families with complex health needs. Other pioneers would put their efforts on reducing the use for hospital care and therefore to ease the economic burden of the local healthcare system. A summary table comparing the key concerns of each pioneer area and the policies implemented to address these issues would be helpful.	We have revised the text in the introduction to make reference to the plans of each pioneer with reference to a summary table contained in the supplementary file which covers the target population, the mechanisms by which they hoped to integrate care, and some examples of specific activities of each Pioneer.

2	1	Introduction	This is related to point 1. After the area-level pioneer plan has been identified, a theoretical framework could be included in the introduction to demonstrate the mechanism through which the outcome (emergency admission rate) is related to which element of the plan. This would help justify the validity of the research question.	This was an observational study and emergency admissions were used as the measure of success because it was a recurrent theme in Pioneer objectives and national policy initiatives. We have outlined the focus on emergency admissions in the bids and made reference to the findings of a previous study which looked at the Pioneers plans in more detail. Each Pioneer was free to propose, develop and attempt to initiate its own horizontal (health and social care) integration strategy and thus there was variation between the Pioneers in what they planned to achieve and how they planned to bring this about. However, most of the Pioneers envisaged that greater emphasis on more seamless multi-disciplinary care outside hospital would improve not just patient experience among older people with multiple long-term conditions but reduce the rate at which such people were admitted to hospital for unplanned care. The validity of the current research question was subsequently enhanced by the finding of our early evaluation of the Pioneer programme (2013-15) which indicated that the Pioneers were giving ever more emphasis in their integration efforts to reducing hospital utilisation, in line with the deteriorating NHS financial position and pressure from NHS England (Erens B, Wistow G, Mounier-Jack S, et al. Early evaluation of Integrated Care and Support Pioneers Programme: Final Report. Policy Innovation Research Unit. 2015. Available: http://piru.lshtm.ac.uk/assets/files/Early_evaluation_of_IC_Pioneers_Final_Report.pdf.
---	---	--------------	---	---

3	1	Methods	Study design. I understand the challenges that one would face when evaluate the impact of a complex health policy, and the DiD analysis could be an effective method. However, the key assumption for DiD analysis to be able to isolate the effect of interest is that the intervention and control units have balanced characteristics in other aspects that would affect the outcome. Therefore, it might be useful to identify matched control areas for each pioneer area based on the factors that would potentially affect people's health needs, and compare the pioneer areas only to the matched control areas (rather than the rest of the country).	Allocating matched control areas for each local authority was considered but we felt that is was not feasible for this analysis. The Pioneers cover a wide variety of different areas (e.g. urban/rural, deprived/not) and are spread across the country and therefore we feel that a broad comparison with the rest of England is relevant. Given the complexity of the health and social care system it would be difficult to identify consistent factors that differentiated the Pioneer and non-Pioneer communities beyond the population level characteristics used in the model. We also run the risk of not finding a suitable match for a number of the Pioneers due to the small number of local authorities available and the requirement for trends to be similar in the pre-intervention period.
4	1	Methods	This is related to point 3. As the study is using HES data, I wonder if the analyses can be applied on hospital-level or even patient-level data, based on the location of hospitals. Then more potential influential characteristics could be controlled for.	The Pioneers were set up around local authorities and CCGs, therefore we don't feel that it would be appropriate to conduct the analysis at the level of the hospital. There is likely to be considerable overlap in some areas between attendees at hospitals who are from both Pioneer and non-Pioneer local areas, in particular in London. We also feel that it would not be appropriate to conduct an analysis at the patient level as we don't know which people within a local authority may have benefited from the Pioneer activities. This is the focus of another piece of work in the evaluation. The goal with this analysis was to look at the area level effect as this is the level that integration is supposed to have occurred at.

5	1	Discussion	This study compared the baseline emergency hospital admission rate with this outcome in 2014/15 and 2015/16, the years immediately after the implementation of the pioneer plan. I wonder if this is a short period for the pioneer plan to show a detectable impact, if there is any. How quickly an impact can be observed would be depending on the focus and delivery of the plan. For example, children with asthma might benefit from the integrated care with reduced risk to be admitted to hospital as emergencies sooner than older population with long-term conditions, for whom we might need longer time to observe a reduced need for emergency hospital care. Therefore, the pre- and post- differences in the outcome as detected in this study might not be taken as evidence of a significant association between the pioneer programme and the use of emergency hospital care.	2015/16 was the limit of the data available to us at the time of analysis. We agree that the explanations of short term effects may be different from longer term effects and are cautious in the discussion. Our emphasis in this analysis is on the range of factors that may be associated with seeking, attaining and initiating Pioneer activity and the impact on emergency admissions in this initial period. Further analyses could look at the longer term.
6	1	Results	Figure 1, as mentioned on page 8 line 26, seems to be missing from the manuscript.	This was included as a separate image file as per the BMJ Open submission instructions.
7	1	Methods/ Results	It would be useful to include more information on the comparison between pioneer and non-pioneer areas for the pre-intervention trend.	We have updated Figure 1 to show monthly data for the pre-intervention trend. Table 1 covers details of the Pioneers and non-Pioneers in terms of population characteristics and deprivation. We have also now statistically examined whether the trends were parallel in the pre-intervention period.

8	2	Intro	For Pioneer groups, the authors state that 'a number aimed to reduce reliance on emergency hospital care'. How many of the 49 had this aim, and should the analysis have been restricted to those which had reducing admissions as an explicit aim? The later focus on reducing emergency hospital use 'as financial austerity bit more deeply' may have come too late for the current analysis (making this group more like second wave Pioneers whom the authors excluded from the analysis). What were the other ones trying to do (granted that the over-arching aim was described as integration)? In general the paper could say more at the start more about what the Pioneers were trying to do (details are given but not till the discussion section) as international readers will have no idea what a 'Pioneer' or 'non-Pioneer' is.	We have revised the text in the introduction and included how many of the pioneers had reducing emergency admissions as an aim or outcome at the outset, supporting information included in the supplementary file. We have also made further reference to work undertaken looking at the detailed plans of the Pioneers. As a sensitivity analysis we also excluded Stoke and North Staffordshire from the regression as this was the one area that made no mention of reducing emergency admissions and had a different focus to the other areas, this is included in the supplementary file. This had some impact on the significance of the results (p value of interaction = 0.0489) but the interaction term to indicate how parallel the trends were over the baseline period was also more significant indicating that they may be less parallel than the overall comparison. We continue to present the results from all wave 1 pioneers as goal of this paper was to look at them as a whole. We have also revised the introduction text to include more detail on the Pioneer program and types of activities they were involved in as well as making reference to existing research. We have moved some of the text from the discussion also to facilitate this.
9	2	Methods	It would be helpful to know how the non-Pioneer groups were selected. Were these all areas in England not defined as Pioneers? This is not clear.	We have added a sentence to the methods which aims to clarify that the non-Pioneers were any local authority not identified as a wave 1 or 2 Pioneer.

10	2	Discussion	The Pioneer programme was not the first in England to promote integration and some previous ones had claimed successes in reducing admissions. Were these excluded from the controls? This would at least be worth a mention in the discussion.	The integrated care pilots are one such previous initiative and in fact several of the Pioneers had previously been integrated care pilots. These local authorities have not been excluded from the Pioneers or controls. This is because the question we were trying to answer in this analysis is whether the Pioneer programme could bring about still greater progress compared to the rest of the country in a health and care system that was generally trying to improve care integration and which had benefited from a series of previous initiatives. We have highlighted that both the Pioneers and non-Pioneers overlapped with integrated care pilots and that this may impact the results.
11	2	Methods	In the analysis, an initial finding is that the intervention group had a 3% greater rate of emergency admissions at baseline (table 1, 10,013 vs 9,705). This raises the possibility that the marginal effect which the authors found in terms of reduced rate of rise in admissions in the intervention group could represent regression to the mean (i.e. if you randomly selected a group with higher rates at baseline, they're likely increase less).	The figure has now been updated to present monthly adjusted rates, the rates are also now adjusted for deprivation which has altered the difference between the Pioneers and non-Pioneers at baseline. The parallel trends analysis also indicated limited bias from differing trends in the baseline period.

12	2	Methods	Rather than using a method that would have used all the data (e.g. segmented regression), the authors have chosen to average the rates in the pre-intervention period and then compare this average with subsequent changes. This seems to depend on their observation that 'the trends were close to parallel' (Results, Trend analysis, line 2) – they may be parallel but they're certainly not flat as, eyeballing figure 1 it's clear that admissions were increasing more rapidly in the Pioneers than non-Pioneers. A statistical opinion on the analytic method chosen should be sought.	Difference in difference looks at two time points, while the model included monthly data for each Pioneer in the pre and post intervention period, with a row of data for each month the model looks at the difference between the two time points overall. The trends were close to parallel in the pre-intervention period. The trends were most similar in 2012/13. This is the baseline year with the least significance for the interaction term to for parallel trends, indicating limited bias ($p=0.9235$) but also the year with the biggest difference-in-differences term (p value 0.0195) when compared to 2014/15. Unlike many forms of segmented regression analysis which look for shifts purely in the intervention group trend, difference in difference enabled the inclusion of a control group.
13	2	Discussion	Other than this, I found the paper well written with judiciously cautious conclusions. One point that could have been made is that some similar evaluations find that the effects of this type of complex intervention take some time to become evident (e.g. ~2 years). Here the reverse seems to be the case, with the effect waning in the second analytic period, though this could have been due to contamination of the control group by other NHS initiatives to reduce admissions.	We have added more detail on this to the discussion and highlighted the contamination of the control group by other initiatives. There was an explicit expectation that other sites would learn from the Pioneers; and, second, many other parts of the country were simultaneously attempting to improve the coordination of their services, often using similar tools and mechanisms of change. Thus one might well expect a pilot programme to show short-run improvements compared with non-pilot areas but that these gains would be relatively short-lived as non-pilot areas learned from the pilots.

14	3	Methods	Pioneer sites were mapped into Local Authorities. I do not find the justification of using local authorities boundaries (instead e.g. CCG boundaries) very compelling. Additionally, CCGs were created in 2013, creating a very serious confounding for the analysis, because of the overlapping with the Pioneer Wave 1.	The mapping of the Pioneers to local authority areas was agreed with each Pioneer site and the local authorities which most closely aligned to the intervention area were used. A population denominator was available for a much longer period for this set of boundaries and it aligns to analysis done in the rest of the evaluation program which used other data including social care which is available at local authority level. As CCG's were implemented across the entire country we do not feel that this should invalidate the analysis as any effect of that change should be seen across both groups.
15	3	Methods	Is there any specific reason of collapsing data at year level (and not e.g. at monthly level)? [line 20,p. 6].	We have revised the presentation of the emergency admissions and the inclusion of the data in the model so that it is at monthly level and included this as a fixed effect.
16	3	Methods	Standardisation: "For the rates, the emergency admissions were directly standardised using the age and sex structure of the 2015 mid-year population estimates for England" [lines 29-33 p 6]. What "directly" means in this context is unclear? Could the authors clarify whether rates were standardised according population estimates for the entire England or according local authority level population estimates?	We have removed references to direct standardisation and revised the text. The age, sex and deprivation population structure for England was used as the reference population for each local authority.
17	3	Methods	I've found the (presentation of the) statistical analysis lacking. In particular: a. Did the author test formally for trend parallelism? "the trend seem close to parallel" [p.8 line 28] is a vague statement.	We have revised the text in this section, presented a graph of monthly data to see the trend over a smaller time period and tested for parallel trends in the pre-intervention period with an interaction term between pioneer status and a monthly time term.

18	3	Methods	b. What is the rationale of using a negative binomial regression model when results are discussed exclusively in terms of emergency admission rates?	Poisson and negative binomial models can be used for count data and include an offset term or exposure variable to account for the population size which means that results can be interpreted in terms of rates. We used a negative binomial model as the count data were over-dispersed which meant that a Poisson model was not suitable.
19	3	Methods	c. "To model the emergency admissions counts as a rate, an offset of the log of the population size in each local authority was incorporated into the model." This is not the standard approach? Were emergency admissions log-scaled?	We have adapted the wording of the Methods in an attempt to make this clearer. Population size was accounted for by the regression model. The offset is used in Poisson/negative binomial regression models to account for exposure. This means that results can be interpreted as rates rather than counts. The variable must be included in log form but does not need to be mentioned in the methods to hopefully make it a clearer read. Emergency admissions were not log scaled prior to modelling.
20	3	Methods	d. I understand "time" takes only 4 values: 1 (2010/11),...,4 (2014/15). Could the authors clarify the reason behind "A baseline of the average of the emergency admissions in the financial years 2010/11 to 2012/13 was created as the period before the initiation of the Pioneer programme in late 2013"? how time enter in the regression analysis? Moreover, Figure 1 reports outcomes from 2007/8, whereas I understand that the statistical analysis uses the "baseline period". Could you also clarify the reasons behind that choice?	Time is entered into the model as a binary variable. Time = 0 represents the pre-intervention baseline period (April 2010 to March 2013), Time =1 represent the post-intervention period either April 2014 to March 2015, or April 2015 to March 2016. We have revised the text at various points in an attempt to make this clearer. The emergency admission data are no longer entered as an average value but ultimately the model treats them as such in the baseline period because the time variable can only take two values in each model (pre/post). We tested several different baseline periods and while 2012/13 showed the biggest effect when compared to 2014/15 (See new supplementary file), it was felt that using the three years prior to the pioneer introduction would allow for comparison of pre intervention differences over a longer time period. We have removed time periods prior to the analysis from the graph in Figure 1.

21	3	Methods	e. "The model accounted for the repeated measures from each local authority and was adjusted for population age group, sex and area-level deprivation" [lines 27-28 p. 7] It is not clear to me whether this would imply the use of a fixed-effect model or instead local authorities dummies were included in the model. Is area-level deprivation a time-invariant characteristic? I am puzzling whether your model is fully identified.	The model used generalised estimating equations to account for the correlation in the individual pioneer local authority emergency admission counts over time. We have revised the inclusion of deprivation so that the model now accounts for any changes in the population structure by age sex and deprivation decile.
22	3	Methods	f. Do you include population size as a regressor in the Poisson specification?	We included population size as the exposure variable for the negative binomial regression model. It is the offset term in the model.
23	3	Results	g. Table 4: what "[.] "calculated" percentage difference" would mean?	We have removed the word "calculated" from the legend of Table 4.
24	3	Discussion	h. I have found the Discussion lacking. I wonder whether the authors could complement the quantitative analysis with some qualitative evidences.	We have included more detail in the introduction around the types of initiatives that the Pioneers introduced and what their focus was. We have also signposted readers to other more detailed work on the activities of the Pioneers.
25	BMJ	Contributor Statement	We have noticed that the initial of "MA" is included in your contributorship statement. However, checking the author list I can't find a name with this initial. Kindly confirm.	We have changed to "MAD".

VERSION 2 – REVIEW

REVIEWER	Marcello Morciano University of Manchester, UK
REVIEW RETURNED	01-Jul-2019

GENERAL COMMENTS	Thank you for a very interesting re-submitted version of this manuscript that I enjoyed reading it. The methodological part is now is clear and complete. The discussion on findings is appropriate and limitations of the study are well identified. The paper has some minor typos that can be easily corrected (e.g. p 3 line 20). Three minor points to be addressed:
---

	p. 6 line 20: “the two time points”. The authors should clarify that they intend the pre- and post-intervention time points (or baseline / follow-up). There is a “Sensitivity Analyses” subsection (p. 8) with results confined in the supplementary material. I wonder whether the authors could just mention somewhere (in that subsection or in the Discussion) what has been learnt from it and whether their main results hold. P 10, line 30: I understand that the overall percentage difference should be -5.73% and not -9.85% as stated in the text.
--	--

VERSION 2 – AUTHOR RESPONSE

Thank you for your email dated 25-Jan-2019 enclosing the reviewer’s comments, which have been very helpful in forming this revision. Our point-by-point responses are given in the accompanying table. Changes to the manuscript are shown with track changes and we have also provided a clean version. We hope that the revised version is stronger, clearer and more helpful.

The author provided a marked copy with additional comments. Please contact the publisher for full details.